# In Search of Molecular Correlates of Fibromyalgia: The Quest for Objective Diagnosis and Effective Treatments

**DOI:** 10.3390/ijms26199762

**Published:** 2025-10-07

**Authors:** Sveva Bonomi, Elisa Oltra, Tiziana Alberio

**Affiliations:** 1Department of Science and High Technology, University of Insubria, Via Manara 7, 21052 Busto Arsizio, VA, Italy; sveva.bonomi@uninsubria.it; 2Escuela de Doctorado, Universidad Católica de Valencia San Vicente Mártir, 46001 Valencia, Spain; 3Department of Pathology, School of Medicine and Health Sciences, Universidad Católica de Valencia San Vicente Mártir, 46001 Valencia, Spain; elisa.oltra@ucv.es

**Keywords:** fibromyalgia, biomarkers, molecular correlates, diagnosis, treatment, drug repurposing, personalized medicine

## Abstract

Fibromyalgia is a chronic syndrome characterized by widespread musculoskeletal pain, fatigue, non-restorative sleep, and cognitive impairment. Its pathogenesis reflects a complex interplay between central and peripheral mechanisms, including altered pain modulation, neuroinflammation, mitochondrial dysfunction, autonomic imbalance, and genetic and epigenetic factors. Evidence from neuroimaging, omics studies, and neurophysiology supports this multifactorial model. Epidemiological updates confirm a global prevalence of 2–8%, with a strong female predominance and a significant impact on quality of life and healthcare costs. Diagnostic criteria have evolved from the 1990 American College of Rheumatology tender points to the 2010/2011 revisions and the 2016 update, improving case ascertainment but still lacking objective biomarkers. Recent omics and systems biology approaches have revealed transcriptional, proteomic, and metabolic signatures that may enable molecularly informed stratification. Therapeutic management remains multidisciplinary, combining pharmacological interventions (e.g., duloxetine, pregabalin, milnacipran) with non-pharmacological strategies such as graded aerobic exercise and cognitive behavioral therapy. Emerging approaches include drug repurposing to target neuroinflammation, mitochondrial dysfunction, and nociceptive pathways. Despite promising advances, progress is limited by small sample sizes, heterogeneous cohorts, and lack of standardization across studies. Future priorities include large-scale validation of biomarkers, integration of multi-omics with clinical phenotyping, and the design of precision-guided trials. By synthesizing mechanistic insights with clinical evidence, this review provides an updated framework for the diagnosis and management of fibromyalgia, highlighting pathways toward biomarker-guided, personalized medicine.

## 1. Introduction

Fibromyalgia (FM) is a chronic syndrome marked by widespread musculoskeletal pain, persistent fatigue, non-restorative sleep, and cognitive impairments (often referred to as *fibro fog*), with sleep disturbance tightly related to the intensity of the pain and affective symptoms [1,2,3,4,5,6,7,8,9,10,11]. The global prevalence is commonly estimated between 2 and 8%, with a clear female predominance and a substantial impact on quality of life, productivity, and healthcare costs [5,6,7,12,13,14,15].

Despite extensive research, FM remains a clinically defined disease (ICD-11, code MG30.01) with heterogeneous presentations and partially overlapping characteristics with other disorders, contributing to diagnostic delay and variable treatment response [1,16,17,18,19,20].

From a nosological perspective, diagnostic criteria have evolved from the 1990 American College of Rheumatology (ACR) tender points to the 2010/2011 revisions and the 2016 update, which incorporate the Widespread Pain Index (WPI), Symptom Severity (SS) scale, and clinical judgment [4,21,22,23,24]. These frameworks have improved standardization and case determination but do not provide objective biomarkers, leaving room for misclassification under phenotypically similar conditions and underscoring the need for molecularly informed stratification [16,17,25,26,27,28].

Clinically, two patterns are recognized: primary, or idiopathic, fibromyalgia, occurring as a stand-alone syndrome without another disorder that fully explains the symptoms, and secondary fibromyalgia, in which the FM symptom complex coexists with or follows other conditions (e.g., inflammatory rheumatic diseases such as rheumatoid arthritis (RA), systemic lupus erythematosus (SLE), or Sjögren syndrome; endocrine/metabolic disorders such as hypothyroidism or diabetes; neuropathies such as chemotherapy-induced or small-fiber neuropathy; and various chronic infections, most commonly hepatitis C virus (HCV), human immunodeficiency virus (HIV), and post-treatment Lyme disease) [3,5,7,12,16,29,30,31].

Differential diagnosis is challenging because several disorders can mimic or overlap with FM—myofascial pain syndrome, osteoarthritis and inflammatory arthritis, hypothyroidism, anemia/iron deficiency, obstructive sleep apnea, major depression/anxiety, myalgic encephalomyelitis/chronic fatigue syndrome (ME/CFS), irritable bowel syndrome (IBS)/functional gastrointestinal disorders, migraine, postural orthostatic tachycardia syndrome (POTS), and small-fiber neuropathy—sharing fatigue, non-restorative sleep, cognitive complaints, and widespread pain [32]. Sleep disorders such as obstructive sleep apnea (OSA) contribute to non-restorative sleep and daytime fatigue, and are confirmed by polysomnography [33]. Endocrine and hematologic conditions (hypothyroidism, iron-deficiency anemia) must also be excluded with targeted tests (TSH/FT4, ferritin/hemoglobin) [34,35]. Autonomic dysfunction, particularly postural orthostatic tachycardia syndrome (POTS), may mimic FM fatigue and cognitive impairment but is distinguished by a heart rate increase >30 bpm on tilt testing [36]. IBS and functional gastrointestinal disorders overlap with FM in abdominal pain and fatigue but can be separated using Rome IV criteria [37]. Psychiatric comorbidities such as major depression and anxiety share fatigue, poor sleep, and concentration difficulties with FM but are identified by core affective syndromes defined in Diagnostic and Statistical Manual of Mental Disorders, Fifth Edition (DSM-5) [38]. A structured differential diagnosis is summarized in Table 1, which contrasts FM with commonly overlapping conditions and lists (i) shared features that can mimic FM and (ii) distinguishing signs and confirmatory tests. Misclassification is favored by the absence of validated biomarkers and the dependence on patient-reported measures; careful exclusion of specific features, targeted laboratory testing, and the application of updated ACR criteria partially help mitigate this risk [21].

In parallel, systems approaches and omics profiling have begun to reveal transcriptional, proteomic, and metabolic signatures that can support diagnosis, prognosis, and therapeutic decision-making [39,40,41,42,43,44,45,46]. Mechanistically, FM is best understood as the convergence of central, peripheral, immune, autonomic, and sleep-related processes [47,48,49,50,51,52,53,54]. Central sensitization, the amplification of nociceptive signaling within the central nervous system (CNS), is supported by neuroimaging evidence of increased responses in the insula, anterior cingulate cortex (ACC), and prefrontal regions during painful stimulation, consistent with a central sensitization framework [55,56,57,58,59,60,61,62].

Peripheral contributions include small-fiber neuropathy—documented in subsets of individuals—and muscular bioenergetic alterations, consistent with mitochondrial dysfunction [63,64,65,66,67,68,69,70,71,72,73]. For example, studies that combined microdialysis and magnetic resonance spectroscopy reported elevated pyruvate with reduced ATP and phosphocreatine in FM compared to matched controls, supporting impaired muscle energy metabolism as a physiological correlate of fatigue and exertional pain [68,74,75]. Low-grade immune activation with altered cytokine profiles (IL-6, IL-8, TNF-α) and neuroinflammation signals in PET have also been described, although the size of the effects and profiles vary between cohorts [48,60,76,77,78,79]. Dysautonomia, including reduced heart rate variability and sympathetic predominance at rest, as well as disturbed sleep architecture (reduced slow-wave sleep and alpha intrusion), is tightly interwoven with pain amplification, cognitive symptoms, and fatigue during the day [49,50,80,81,82,83].

Management of FM requires a multidisciplinary approach. Pharmacological options such as duloxetine, pregabalin, and milnacipran can alleviate pain and associated symptoms in a proportion of individuals; non-pharmacological therapies, including graded aerobic exercise and cognitive behavioral therapy (CBT), are core elements of care [12,84,85,86,87]. Emerging strategies leverage drug repurposing to target neuroinflammation, mitochondrial bioenergetics, or nociceptive pathways with agents that have established safety profiles (e.g., low-dose naltrexone and sublingual cyclobenzaprine), allowing faster clinical translation while highlighting the need for adequately powered and mechanistically anchored trials [62,88,89,90,91]. In addition, computational pipelines using transcriptomic signature matching and network pharmacology broaden the search space for candidate therapies [90,91,92,93].

In this review, we integrate clinical, neurobiological, and molecular evidence to outline an updated framework for FM. Specifically, we (i) summarize epidemiology and diagnostic criteria with an emphasis on strengths and current limitations; (ii) synthesize mechanistic insights across central, peripheral, immune, autonomic, and sleep axes; and (iii) evaluate pharmacological, non-pharmacological, and repurposed interventions in light of emerging biological knowledge. Although candidate gene studies have suggested associations with FM [94], these require replication in larger cohorts; more recent multi-omics approaches provide new directions for biomarker discovery [39,44,46]. Throughout, we identify key gaps (small sample sizes, cohort heterogeneity, and limited standardization) and propose priorities for biomarker validation and precision medicine [1,20,39,40,41,47,70,76,95,96].

A structured overview of these aspects is provided in Table 2, which condenses the main clinical characteristics, pathophysiological axes, diagnostic frameworks, therapeutic approaches, and future research directions in FM.

## 2. Pathogenetic Mechanisms and Molecular Correlates

The pathogenesis of FM reflects a dynamic interplay between alterations in the central and peripheral nervous systems, mitochondrial and metabolic dysfunction, low-grade inflammation, autonomic imbalance, and sleep disruption. These axes are variably expressed between patients and likely interact over time, helping explain clinical heterogeneity and the inconsistent effect sizes observed for single-target interventions [3,47,50]. In the sections that follow, we synthesize evidence across these domains into a coherent, clinically oriented narrative, integrating mechanistic detail and grounding each assertion in the primary literature. An integrative summary of the major axes, representative findings, and clinical correlates is provided in Table 3.

### 2.1. Skeletal Muscle and Mitochondrial Bioenergetics

Muscle involvement in FM has long been investigated as a potential contributor to exercise intolerance, post-exertional symptom exacerbation, and localized pain. Early morphological studies reported limited or non-specific histopathology [23,25], yet subsequent work identified ultrastructural changes consistent with mitochondrial distress and elevated oxidative stress in subsets of patients [65,66]. Converging metabolomic and physiology studies support impaired high-energy phosphate handling and bioenergetic inefficiency during and after exertion, with microdialysis and magnetic resonance spectroscopy indicating altered glycolytic intermediates and reduced phosphocreatine (PCr) recovery [51,68]. These findings are complemented by reports of reduced coenzyme Q_10_ and antioxidant defense, including tissue-level evidence from skin biopsy studies [26], as well as increased lipid peroxidation and nitrosative stress in muscle or blood [13,42,43,67,97], which indicates a redox imbalance that may bias muscle toward fatigue and nociceptive sensitization.

At the molecular level, altered redox buffering and mitochondrial function typically involve enzymes such as superoxide dismutase (SOD), glutathione peroxidase (GPX), and catalase (CAT); accumulation of peroxidation products (malondialdehyde [MDA], 4-HNE); and impaired ATP/PCr kinetics and electron transport chain vulnerability at respiratory chain complexes (I–IV), with downstream effects on the handling of pyruvate/lactate and the oxidation of fatty acids (e.g., carnitine shuttle activity). Dysregulation of biogenesis regulators (PGC-1α/NRF1/TFAM) provides a plausible transcriptional route to sustained mitochondrial inefficiency. This aligns with the mitochondrial–redox imbalance described in FM [26,52].

Together, the data support a model in which peripheral metabolic stress acts as a sustained afferent drive that can reinforce hyperexcitability in vulnerable individuals [66,68]. Complementary observations of central biofluids, including altered amino acid profiles and neurochemical shifts in cerebrospinal fluid (CSF), further fit a bioenergetic–metabolic framework for central excitability [98,99].

### 2.2. Dorsal Root Ganglia and Small-Fiber Pathology

Peripheral sensory neurons located in the dorsal root ganglia (DRG) and their distal terminals may sustain peripheral sensitization and fluctuation of symptoms, positioning the DRG as a plausible “pain factory” in FM [71]. Skin biopsy and corneal confocal microscopy studies document reduced intraepidermal nerve-fiber density or morphological abnormalities in a subset of patients, consistent with small-fiber neuropathy [63,64,73,100]. These changes align with clinical features such as burning pain and allodynia. Experimental and translational evidence further shows that inflammatory signaling within DRG can sensitize primary afferents and amplify the nociceptive input to the CNS [82,101], providing a plausible substrate to maintain widespread pain in the absence of overt large-fiber neuropathy.

Key molecular mediators of peripheral hyperexcitability at the nociceptor level include voltage-gated sodium channels Nav1.7/Nav1.8, TRP channels (TRPV1/TRPA1), purinergic P2X3 receptors, and neurotrophin signaling (NGF/TrkA), with cytokine/chemokine cues (e.g., TNF-α, IL-6, CCL2) and neuropeptides (substance P, CGRP) sustaining sensitization and neurogenic inflammation at the terminal and DRG soma. Although standardized diagnostic protocols for small-fiber involvement are still evolving, the DRG/small-fiber axis offers a mechanistic bridge between peripheral triggers and central sensitization [63,64,73].

### 2.3. Sensitization and Pain Network Plasticity

A large body of neuroimaging and psychophysical evidence has driven the central sensitization theory as a key driver of FM symptomatology. Functional magnetic resonance imaging (fMRI) and related modalities reveal enhanced responses to painful and, at times, harmless stimuli within the insula, anterior cingulate cortex, thalamus, and prefrontal regions [9,57,62,92]. Altered resting-state connectivity within pain modulatory networks has also been reported [58,60], consistent with plastic changes in network organization. Complementary theoretical and experimental frameworks describe a lowered threshold for nociceptive transmission and impaired descending inhibition [14,19,24], consistent with canonical models of central sensitization. These observations are consistent with the clinical effects of agents that improve inhibitory monoaminergic tone in selected patients [3], although interindividual variability remains substantial. Molecular imaging and CSF-level studies provide convergent support for central excitability and neuroinflammatory mechanisms in vivo [99,102,103].

In particular, CSF and neurochemical assays point to elevated substance P and altered monoamine handling (including platelet serotonin uptake and metabolite balance), while amino acid patterns such as the branched-chain amino acids–tryptophan ratio suggest shifts in serotonergic and nitric oxide pathways. During experimentally induced sensitization (temporal summation/“wind-up”), proton MR spectroscopy and microdialysis detect elevated insular glutamate, which is reduced by gabapentinoids. Translocator protein positron emission tomography (TSPO-PET) concurrently indicates glial activation (microglia/astroglia) in cingulate, insula, and thalamus—molecular correlates of sustained central hyperexcitability and neuroinflammation. Within central circuits, glutamatergic receptors (NMDA/AMPA; metabotropic mGluR5), GABAergic tone, and monoaminergic modulators (serotonin transporter [SERT/SLC6A4], norepinephrine transporter [NET/SLC6A2]) shape the gain of nociceptive transmission; activity-dependent trophic signals (BDNF/TrkB) and microglial mediators (e.g., IL-1β, IL-6, COX-/PGE2) contribute to synaptic potentiation and disinhibition that characterize central sensitization.

### 2.4. Immune Signaling, Low-Grade Inflammation, and Symptom Modulation

Immune profiling studies report patterns compatible with low-grade systemic inflammation in a proportion of FM patients, including altered levels of interleukins and chemokines that correlate with pain and fatigue in some cohorts. Most signals have been detected in peripheral blood (serum/plasma) using multiplex bead-based immunoassays or ELISAs, with several studies extending the same panels to CSF, a minority of which have used discovery proteomics on the same matrices. In all these specimens, the most reproducible differences are modest increases in IL-6, IL-8/CXCL8, TNF-α, and MCP-1/CCL2, sometimes accompanied by acute-phase reactants (e.g., C-reactive protein (CPR)/serum amyloid protein A (SAA)); where reported, concentrations scale with intensity of pain, fatigue, and sleep disturbance, consistent with neuroimmune coupling [48,76,78]. Neuroimmune crosstalk is further supported by TSPO-PET evidence of increased glial activation in pain-related brain regions [60], which aligns peripheral cytokine signals with central neuroinflammation and heightened sensory gain. Heterogeneity across studies underscores the likelihood of immunologically defined subgroups and highlights the need for standardized sampling (matrix, timing, pre-analytics), careful phenotyping, and longitudinal designs [76,78]. Epigenetic work in blood (including sibling designs) strengthens causal inference for immune–stress axes that influence pain processing [104,105]. Taken together, a compact multi-analyte panel centered on IL-6, IL-8/CXCL8, TNF-α, and MCP-1/CCL2 (with CRP/SAA as an acute-phase context) emerges as a pragmatic biomarker candidate set for prospective validation [76,78].

### 2.5. Autonomic Dysregulation and Interoceptive Burden

Autonomic nervous system (ANS) imbalance is frequently observed in FM, with reports of reduced heart rate variability, sympathetic predominance at rest, and exaggerated cardiovascular responses to orthostatic or psychosocial stressors [49]. This imbalance aggravates pain and fatigue through three coupled, mechanistically traceable routes: (i) when baroreflex efficiency is low, the heartbeat-linked antinociceptive “brake” weakens, so descending inhibition falls—an axis that functionally depends on adrenergic signaling (β1/β2- and α2A-adrenergic receptors), catecholamine clearance (NET; COMT; MAO-A/B), and brainstem circuitry; (ii) persistent sympathetic tone and microvascular dysregulation reduce muscle perfusion and favor metabolite accumulation, increasing peripheral nociceptive input—here, endothelial/vascular mediators such as eNOS (NOS3), endothelin-1 (EDN1), and prostacyclin synthase (PTGIS), together with neuropeptides substance P and CGRP, shape local flow and excitability; and (iii) heightened interoceptive signaling from cardiovascular/visceral sources makes bodily sensations highly salient, amplifying symptom perception and effort costs. These processes reinforce global sensitization and interact bidirectionally with small-fiber pathology and local inflammatory milieus described above [49,50], as summarized in Table 3.

Integrative models of pain neurobiology place the ANS inside nociceptive control loops: baroreflex arcs and the vagus-mediated cholinergic anti-inflammatory pathway (α7 nicotinic acetylcholine receptor on immune cells) converge on brainstem pain-inhibitory hubs (periaqueductal gray–rostral ventromedial medulla) and limbic salience networks (insula, anterior cingulate). When these loops are off-balance—reduced baroreflex gain, low vagal tone—descending inhibition weakens and inflammatory signaling increases, while salience assignment to bodily signals rises; the net effect is a self-sustaining state of hyperalgesia and fatigue that also couples with sleep disruption and affective load in chronic pain states [106,107].

### 2.6. Sleep Architecture and Bidirectional Symptom Escalation

Disturbances in sleep continuity and architecture—reduced slow-wave sleep, alpha intrusion, and fragmentation—are consistently reported in FM, associated with the intensity of pain the next day and cognitive symptoms [80,81]. Persistent sleep problems prospectively predict incident FM in population cohorts [83], providing prospective evidence of a causal link. Experimental models of partial sleep deprivation induce hyperalgesic states and diffuse tenderness in otherwise healthy individuals [50,108], supporting a pathway in which non-restorative sleep lowers pain thresholds and perpetuates symptom cycles. Therefore, sleep-focused interventions play an important role in comprehensive management strategies and can increase the efficacy of mechanistically targeted treatments [50]. Neurochemical studies of the CSF and central imaging findings complement this sleep–pain amplification model [99,102]. On the molecular side, sleep–pain coupling is consistent with altered GABAergic tone, adenosine signaling (A1/A2A), and the serotonergic/tryptophan–melatonin axis (5-HT/SERT and pineal melatonin output), which jointly modulate sleep depth, arousal threshold, and nociceptive gain; convergence with cytokine rhythms (e.g., IL-6, TNF-α) provides an immune–sleep route by which non-restorative sleep can amplify pain sensitivity [50,99,108].

## 3. Omics Approaches in Fibromyalgia

Omics technologies enable the simultaneous assay of thousands of features and the modeling of their multivariate structure (co-expression/co-abundance and pathway co-regulation). Correlation networks and latent-factor models compress high-dimensional signals into biologically coherent modules linked to pain, fatigue, sleep, and autonomic measures, enabling a mechanism- and endotype-oriented view of FM.

In this section, we detail what has been measured (layers, tissues, and technologies) and which recurrent molecular patterns have emerged, highlighting concrete molecular handles that align with the pathogenetic axes in Section 2 [1,39,44].

### 3.1. Genomics, Epigenetics, and Transcriptomics

Genetic studies have implicated variants in neurotransmission and stress–response pathways. Polymorphisms in *SLC6A4* and *GRIA4* are linked to glutamatergic signaling, functional variants in *COMT* alter catecholamine metabolism and pain sensitivity, and allelic variation in *SLC6A4* (5-HTT) influences serotonergic tone. These associations suggest that inherited predisposition shapes sensory amplification, autonomic imbalance, and affective comorbidities [39]. Polygenic analyses further indicate shared risk with chronic pain syndromes and affective disorders, supporting a polygenic and network-based rather than monogenic architecture.

Activation of human endogenous retroviruses (HERVs) in PBMCs has been reported to discriminate FM, ME/CFS, co-diagnosed, and healthy controls, with specific HERV–immune gene modules correlating with symptom severity [109]. Since HERV expression is normally suppressed by epigenetic regulators such as SETDB1 and TRIM28, these findings suggest that loss of retroviral silencing may contribute to immune dysregulation in FM.

Epigenetic studies further highlight regulatory layers that bridge genetic susceptibility with the environment and disease expression. Differential DNA methylation at immune and oxidative-stress loci parallels transcriptomic modules [39], supporting a role for stable epigenetic programming of immune tone and redox balance. Hypermethylation at the *GRM2* locus has been reported in sibling-controlled cohorts and identified as a central hub in methylation networks, suggesting a protective effect against FM [39].

MicroRNA profiling across blood, PBMCs, CSF, and skin implicates multiple regulatory nodes: the let-7 family is linked to nociceptor excitability and immune tone, miR-21-5p to inflammation and TGF-β-related remodeling, miR-146a-5p and miR-155-5p to NF-κB and JAK–STAT feedback, miR-27a/b and miR-223-3p to lipid metabolism–myeloid activation, miR-103/107 and miR-320a to insulin signaling and mitochondrial coupling, and neuronal miR-132/212, miR-124-3p, and miR-9-5p to synaptic plasticity. Increased miR-34a-5p has been associated with mitochondrial stress and apoptosis. These regulatory layers converge on immune–metabolic and neuroplastic axes enriched in NF-κB/JAK–STAT signaling, glutamate–GABA balance, monoamine transport, and OXPHOS/redox biology [39,45,110].

Overall, genetic polymorphisms provide stable risk markers, whereas epigenetic mechanisms—including DNA methylation, miRNAs, and HERV activation—capture dynamic adaptations to stress, environment, and disease progression. Their integration with proteomic and metabolomic layers is key for personalized medicine in FM. A structured overview of these findings is provided in Table 4.

### 3.2. Proteomics and Metabolomics

Proteomics in FM has primarily assayed serum or plasma and, in several studies, cerebrospinal fluid, with occasional saliva datasets, using multiplex immunoassays or ELISAs and, less frequently, discovery proteomics. The most reproducible differences include modest but consistent increases in IL-6, IL-8/CXCL8, TNF-α, and MCP-1/CCL2, sometimes accompanied by CRP or SAA as an acute-phase context. Pathway enrichment maps these panels to leukocyte chemotaxis and trafficking, cytokine signaling through JAK/STAT and NF-κB, coagulation and complement cascades, and lipid signaling, coherently linking peripheral immune tone with central neuroinflammation and glial activation [1].

Metabolomic profiling, both targeted and untargeted, has primarily interrogated serum or plasma and urine, with muscle microdialysis used to sample the interstitial milieu during or after exertion. Across matrices, recurrent alterations involve amino acid metabolism, including the branched-chain amino acids, tryptophan with the serotonin–kynurenine–melatonin axis, and glutamate and GABA; bioenergetic intermediates from glycolysis and the tricarboxylic-acid cycle, such as lactate, pyruvate, citrate, and succinate; and lipid classes that index mitochondrial flux and membrane remodeling, including short-, medium-, and long-chain acylcarnitines, lysophosphatidylcholines and phosphatidylcholines, and sphingomyelins and ceramides [44,46]. At the pathway level, targeted and discovery workflows show concordant enrichment for mitochondrial beta-oxidation and OXPHOS with redox imbalance, lipid–amino acid crosstalk, and modulation of tryptophan fate. These features coherently link peripheral bioenergetic stress to the central excitability and neuroinflammatory signals discussed in Section 2 [1,44,46].

### 3.3. Analytical Approaches for Module Discovery and Endotyping

To move from long lists of molecules to interpretable biology, set-level methods such as Gene Set Enrichment Analysis (GSEA) and Over-Representation Analysis (ORA) test whether entire pathways or gene sets are coordinately altered [111,112]. Correlation networks and latent-factor models yield modules that map onto clinical domains (e.g., cytokine-rich modules → fatigue/sleep; acylcarnitine/OXPHOS → exertional intolerance; tryptophan–serotonin–kynurenine → pain/sleep), defining endotypes for stratification and biomarker-guided interventions.

Moving from univariate contrasts to modules and pathways increases biological interpretability and translational potential, leading to compact multi-analyte panels that can support mechanism-anchored therapeutic strategies (Figure 1).

Despite convergence across single omic layers, FM still lacks rigorous, joint multi-omics integration across independent cohorts. For multi-system conditions, combining genomics, epigenomics, and transcriptomics with proteo–metabolomics and deep phenotyping should improve the signal-to-noise ratio, reveal clinically meaningful endotypes, and accelerate biomarker discovery and mechanism-anchored repurposing [90,91,111,112].

## 4. Therapeutic Strategies and Drug Repurposing in Fibromyalgia

Guideline care combines patient education, optimization of sleep and activity, graded physical exercise, and psychological or behavioral therapies with selected pharmacological options [12,27,113]. Non-pharmacological strategies such as aerobic and resistance training, sleep hygiene, and cognitive or acceptance-based therapies remain foundational and often potentiate the effects of medication by reducing nociceptive gain and stabilizing sleep–autonomic axes. A structured overview of available therapeutic options, including approved pharmacological agents, core non-pharmacological care, emerging repurposed agents, and future perspectives, is provided in Table 5.

Approved pharmacotherapies historically entered FM care through a mixture of symptom analogy and partial alignment with central pain biology. Serotonin–norepinephrine reuptake inhibitors such as duloxetine and milnacipran increase synaptic monoamines and thereby augment descending inhibition along periaqueductal gray–rostral ventromedial medulla pathways while also modulating affect and sleep [3,27]. This pharmacology maps to molecular observations of altered monoamine handling, including platelet serotonin transporter uptake and changes in monoamine metabolites, and to the clinical coupling between mood, sleep, and pain thresholds. Gabapentinoids such as pregabalin bind to the $\alpha_{2}\delta$ subunit of voltage-gated calcium channels and reduce presynaptic release of glutamate and other transmitters; this aligns with magnetic resonance spectroscopy and microdialysis evidence of elevated insular glutamate and its partial normalization after treatment [3,103]. Low-dose tricyclics or cyclobenzaprine provide mixed noradrenergic and serotonergic reuptake inhibition together with antihistaminergic and sedative effects that can improve sleep continuity and pain in a subset of patients [12,27]. In contrast, chronic opioid therapy is discouraged given limited efficacy for centralized pain, risk of hyperalgesia, and adverse outcomes [12,113]. Across agents, average effect sizes remain modest and heterogeneous, reinforcing the shift from symptom analogies to mechanism-based stratification.

Pharmacological options such as duloxetine, pregabalin, and milnacipran can alleviate pain and associated symptoms in a subset of individuals; non-pharmacological therapies, including graded aerobic exercise and cognitive behavioral or acceptance-based therapies, are core elements of care [12,84,85,86,87].

### 4.1. Emerging and Adjunctive Approaches: Where They Come From and What They Target

Several adjunctive or investigational therapies arise directly from the molecular correlates outlined in Section 2 and Section 3. Low-dose naltrexone has been proposed to attenuate microglial activation via Toll-like receptor 4 antagonism and to modulate endogenous opioid tone. The rationale builds on in vivo evidence of neuroinflammation and glial activation on TSPO-PET, together with CSF neuropeptide and monoamine abnormalities. Small randomized and single-center studies suggest benefit in subsets, although larger biomarker-stratified trials are needed [88,114]. Ketamine, a non-competitive NMDA receptor antagonist, emerged from laboratory evidence of glutamatergic hyperexcitability: spectroscopy and microdialysis studies show increased insular glutamate during nociceptive gain states, and NMDA blockade can transiently reduce central sensitization [103,115]. Clinical responses are often short-lived, motivating biomarker-guided selection and maintenance strategies. Cannabinoid-based medicines such as nabilone engage CB1 and CB2 receptors that regulate nociception, sleep, and immune tone; trials indicate modest effects on sleep and pain with tolerability considerations [116,117]. Within monoamine pathways, choosing between SNRIs and low-dose tricyclics can be anchored to peripheral proxies of central tone, such as platelet serotonin transporter uptake or monoamine metabolite balance, and to the degree of coupling between mood, sleep, and pain [3]. Finally, given the evidence for reduced baroreflex gain and low vagal tone in subsets, strategies that raise vagal activity and stabilize autonomic balance—ranging from aerobic training and behavioral sleep consolidation to selected pharmacological adjuncts—may secondarily reduce nociceptive gain and fatigue by acting on baroreflex and cholinergic anti-inflammatory reflexes and by moderating adrenergic tone [12,50].

### 4.2. Mechanism-Anchored Repurposing: Linking Molecules, Pathways, and Drugs

Data-driven repurposing methods (signature matching, network medicine) can align dysregulated FM modules with plausible drug mechanisms (Figure 1) [90,91,93]. Neuroinflammation and glial activation, reflected by cytokine panels, including IL-6, IL-8/CXCL8, TNF-α, and MCP-1/CCL2, and by in vivo neuroinflammatory imaging, suggest agents that modulate microglial activation or downstream cytokine signaling, exemplified by Toll-like receptor 4 modulation with low-dose naltrexone [118]. Glutamatergic hyperexcitability, indicated by insular glutamate elevations and wind-up phenomena, motivates NMDA antagonism with ketamine and reduction of presynaptic glutamate release with pregabalin via the $\alpha_{2}\delta$ subunit. Monoamine imbalance, inferred from transporter proxies and metabolite shifts, supports the use of SNRIs and related agents to augment descending inhibition. Mitochondrial and redox–lipid metabolism signals, including acylcarnitine patterns and oxidative phosphorylation or redox transcripts and microRNAs, motivate exploration of metabolic and redox support aligned to bioenergetic stress signatures. The general principle is to select agents that intersect dysregulated nodes demonstrated in vivo rather than by symptom resemblance.

### 4.3. Future Directions: From Panels to Trials

A practical path forward begins with compact multi-analyte panels anchored in the convergent axes described above, optimized for matrix, pre-analytics, and longitudinal stability [1]. Recent summaries emphasize an integrated gene–environment perspective and incremental validation pathways for biomarker-guided, stratified trials [114,119,120,121]. These panels should support endotype definition using correlation networks and latent factors that yield neuroinflammatory or glial, glutamatergic, monoaminergic, and mitochondrial or redox profiles tied to clinical domains like pain, sleep, fatigue, and autonomic measures. Pathway-constrained signature matching can then nominate candidate interventions for each endotype, avoiding choices based solely on symptom resemblance [90,91] (Figure 1). The central test of this framework is a sequence of biomarker-guided adaptive trials that randomize within endotypes and include early pharmacodynamic readouts [92,93]. In summary, current therapies largely originated from symptom analogies and broad central pain biology, whereas future progress is likely to depend on anchoring drug selection to measurable molecular correlates and on testing this paradigm in stratified clinical trials.

## 5. Discussion

Fibromyalgia emerges as a paradigmatic example of a complex systems-level disorder in which central pain amplification, peripheral bioenergetic stress, immune–metabolic dysregulation, and disturbances of sleep and autonomic tone generate a multidimensional clinical picture. The need for improved FM management arises from its high global burden, the substantial impact on quality of life and productivity, and healthcare system utilization [1,5,6,7,12,14,15,122,123]. Despite more than three decades since the original ACR classification criteria [4], diagnosis remains clinical and often delayed, reflecting the heterogeneity of symptoms and the lack of objective biomarkers [16,17,25,29,124,125,126].

Although FM shows a clear female predominance at the population level, more recent applications of updated diagnostic criteria suggest a reduced female-to-male ratio (historically ∼9:1, now closer to ∼3:1). This shift may reflect both biological sex-specific mechanisms (hormonal, neuroimmune, nociceptive) and healthcare access biases. Reporting sex-stratified prevalence, presentation, and treatment outcomes should therefore be standard in future trials [114,119].

FM cannot be reduced to a single pathway. Functional neuroimaging and neurophysiology show enhanced responses to nociceptive and non-nociceptive stimuli, altered connectivity of the resting state, and impaired control of descending inhibitory pain-modulatory networks [19,24,47,56,57,58,59,62,92,102]. These data are consistent with the central sensitization model [3,14,19,50], pointing to glutamatergic receptor activity, reduced GABAergic tone, monoamine transporter imbalance, and microglial mediators as plausible drivers of sustained central hyperexcitability.

Peripheral drivers are also evident. Mitochondrial dysfunction and oxidative stress in skeletal muscle plausibly contribute to exercise intolerance and post-exertional exacerbation [13,43,48,51,65,68,127]. Small-fiber pathology in a subset of patients provides structural correlates for altered nociception [63,64,71,73,101]. Sleep fragmentation and autonomic imbalance further amplify central and peripheral dysfunction [49,50,80,81,83,84,108,128]. Reduced baroreflex gain and low vagal tone weaken descending antinociception and the cholinergic anti-inflammatory reflex; persistent sympathetic drive and microvascular dysregulation—implicating eNOS (NOS3), endothelin-1 (EDN1), prostacyclin synthase (PTGIS), and neurovascular peptides—favor metabolite accumulation and heightened interoceptive salience, providing a route by which autonomic tone sustains hyperalgesia and fatigue.

Together, these loops create a dynamic system in which pain, fatigue, cognitive dysfunction, and sleep disturbance reinforce each other, accounting for the daily fluctuations of symptom severity and the difficulty of stabilizing long-term treatment response.

These multilayered interactions explain clinical heterogeneity and variable treatment response [22,54,124,129]. Molecular and genetic insights reinforce a network view of FM. Candidate and pathway-focused genetic studies implicate monoaminergic and catecholaminergic signaling and ion-channel biology—findings more consistent with modulation of pain processing than simple case–control risk—across loci that include serotonin and norepinephrine transporters (SLC6A4, SLC6A2), catecholamine metabolism (COMT, MAO-A/B), serotonergic receptors (HTR2A), dopaminergic variation (DRD4), and neurotrophin pathways (BDNF) [75,77,94,130,131,132,133,134,135,136,137,138,139,140].

Epigenetic and retroviral findings (DNA methylation at immune–redox loci, HERV derepression) further integrate environmental exposures with immune and stress signaling, suggesting layered regulatory mechanisms relevant to FM expression [39,109].

Transcriptomic analyses in PBMCs reveal coordinated perturbations that span immune pathways (e.g., Th17/type I interferon signatures) and long non-coding RNA networks that regulate hub genes [39,111,141,142,143]. In line with this, miRNA studies demonstrate context-specific dysregulation among biofluids and tissues—CSF, skin, PBMCs—linking neuronal excitability, immune tone, and metabolic adaptation to symptoms [40,101,144,145,146,147]. Rather than large, isolated effects, coherence emerges at the pathway and module levels, which is precisely the scale at which integrative methods such as gene set enrichment operate effectively [93,111,112].

A summary of molecular correlates and related pathogenetic axes is presented in Figure 2.

Toward mechanistically anchored biomarkers, these converging molecular signals motivate a shift from single-analyte predictors to composite signatures that integrate multiple layers (genomic/epigenomic, proteomic/metabolomic) with physiology (sleep–autonomic metrics) and clinical phenotypes. In practice, this implies standardized pre-analytics, explicit matrix selection, harmonized pipelines for data integration, and external validation across heterogeneous cohorts [90,91,148]. In parallel, mechanism-guided drug discovery and repurposing can exploit such signatures: examples include stratification of low-dose naltrexone or glutamatergic modulators based on immune or metabolomic fingerprints, and prioritizing candidates through systematic repurposing frameworks [88,90,91,115,116,117,148,149].

In practical terms, endotype panels might include (i) a neuroinflammatory module (IL-6, IL-8/CXCL8, TNF-α, MCP-1/CCL2 ± CRP/SAA); (ii) a bioenergetic–redox/lipid module (acylcarnitines, lactate/pyruvate, OXPHOS-related metabolites); (iii) a neurotransmission module (glutamate/GABA; monoamine proxies); and (iv) autonomic–sleep metrics (HRV components, baroreflex sensitivity, slow-wave sleep proportion). These map onto the pathogenetic axes summarized in Table 3 and can be tracked longitudinally for within-person dynamics.

The omics evidence reviewed here aligns with this trajectory. Proteomic and metabolomic studies in serum/plasma, urine, saliva, and CSF converge on pathways related to mitochondrial stress, redox imbalance, lipid and amino acid metabolism, and low-grade inflammation, often correlating with symptom domains [1,44,46,66,78,98,150,151,152,153,154,155]. Again, pathway-level consistency outweighs single-marker effects, arguing for multi-feature panels subject to rigorous external validation and continuous performance monitoring [39,44,45,46]. Critically, several of these panels are built from concrete molecules—IL-6, IL-8/CXCL8, TNF-α, MCP-1/CCL2, and CRP/SAA on the protein side; branched-chain amino acids, tryptophan/kynurenine balance, glutamate/GABA, lactate/pyruvate, acylcarnitines, and phospholipids on the metabolite side—which provide assayable handles for translation and for aligning peripheral readouts with central neurochemical and imaging findings.

Consistency at the pathway (not single-molecule) scale argues for compact multiplex assays with calibration/quality control across sites and predefined drift monitoring to ensure clinical robustness.

Therapeutically, a person-centered multimodal model remains the standard of care. SNRIs and pregabalin produce modest average benefits with substantial interindividual variability [27,84,113,156,157,158], while structured exercise, CBT/acceptance-based approaches, tai chi, and acupuncture provide clinically significant improvements with favorable safety [12,73,85,86,87,159,160,161]. Emerging and repurposed agents—such as low-dose naltrexone and N-methyl-D-aspartate antagonists with carefully controlled protocols—illustrate the potential of mechanism-guided innovation but demand larger randomized, biomarker-stratified trials to quantify effect heterogeneity [88,89,115,162]. In this context, composite biomarker panels can reduce trial and error by enabling earlier identification of likely responders [90,91,163].

Important limitations temper current inferences. Many mechanistic and omics studies are cross-sectional and underpowered; case definitions, outcome measures, and comorbidity burdens vary widely; and pre-analytical and analytical heterogeneity complicates the picture [16,17,126,129]. Most importantly, FM is dynamic. Symptom trajectories and molecular signatures likely fluctuate over time. Longitudinal, harmonized cohorts that integrate deep phenotyping, multi-omics, and digital monitoring of sleep, activity, and autonomic function are needed to model within-person dynamics and delineate mechanistic subtypes [60,69,70,71,76,78].

In a broader perspective, FM exemplifies the challenge of chronic complex disorders that defy linear pathophysiological narratives. Network medicine offers both a conceptual framework and practical tools for unifying high-dimensional datasets into clinically actionable models. Moving forward, critical priorities include (i) harmonization of diagnostic criteria and outcome measures across studies, (ii) integration of multi-omics data with deep phenotyping to identify reproducible biomarkers, and (iii) implementation of large, multicenter trials that stratify patients according to biological and clinical signatures rather than treating FM as a homogeneous entity. Such steps are essential to overcome the current “one-size-fits-all” approach and to lay the groundwork for biomarker-guided, precision medicine strategies in FM.

Finally, while the proliferation of candidate mechanisms highlights the complexity of FM, it also risks fragmenting the field. A coordinated effort to build shared datasets, apply standardized bioinformatic pipelines, and validate promising biomarkers across independent populations will be key to moving from descriptive correlations to actionable therapeutic targets. To prevent field fragmentation, coordinated consortia should align discovery, assay translation, and interventional testing through shared governance, versioned protocols, and rolling external validation waves.

In conclusion, FM should not be conceptualized as a disorder of a single pathway but as an emergent property of dysregulated interactions between neural, immune, metabolic, and autonomic systems. This model explains the modest efficacy of single-target treatments, legitimizes multimodal care, and supports biomarker-guided, mechanism-based clinical trials. By bringing molecular correlates to the center of diagnosis and therapy, the field can move from a clinically defined syndrome to a biologically grounded condition, enabling earlier diagnosis, more personalized interventions, and improved outcomes.

## Figures and Tables

**Figure 1 ijms-26-09762-f001:**
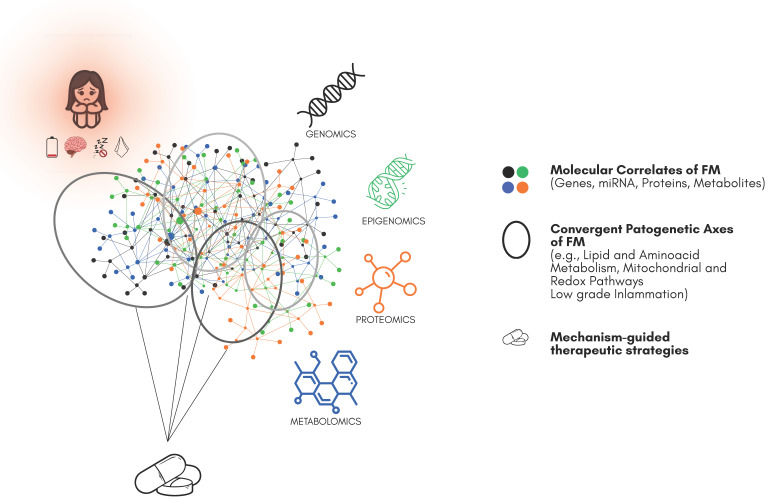
Omics-based technologies, including genomics, transcriptomics, epigenomics, proteomics, and metabolomics, provide a comprehensive view of the molecular alterations associated with fibromyalgia, enabling the identification of candidate biomarkers and disease-related factors. By integrating these data, systems biology approaches can uncover commonly dysregulated pathways and molecular networks that underlie disease mechanisms. Such knowledge can inform pharmacological strategies by guiding the development of targeted therapies or the repurposing of existing drugs, ultimately allowing interventions to be directed toward specific molecular targets or dysfunctional pathways identified through these molecular studies.

**Figure 2 ijms-26-09762-f002:**
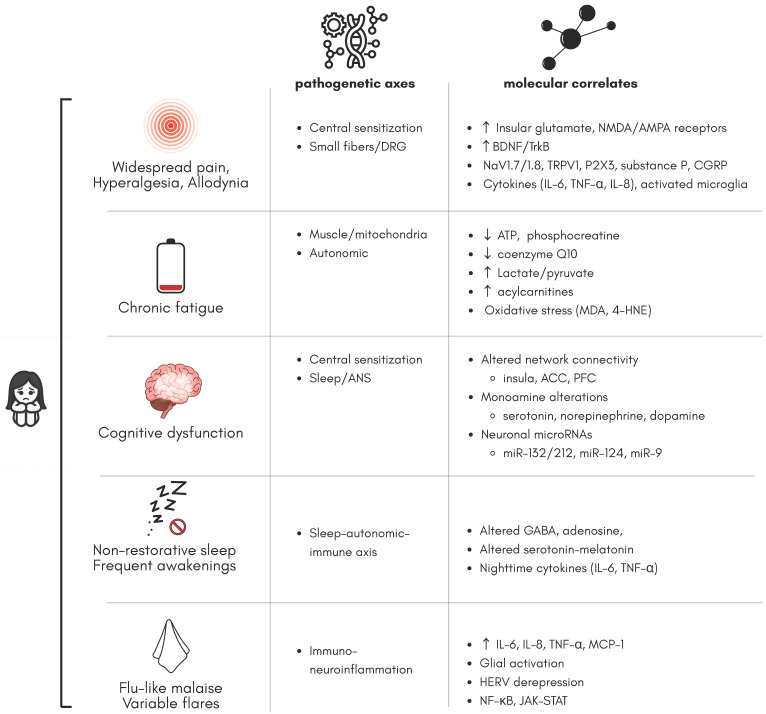
Summary of key symptoms with pathogenetic axes and associated molecular findings.

**Table 1 ijms-26-09762-t001:** Differential diagnosis of fibromyalgia versus overlapping conditions: shared and distinguishing features.

Condition	Overlap with FM	Distinguishing Features
ME/CFS	Fatigue, unrefreshing sleep Cognitive complaints	Post-exertional malaise ≥ 24–48 h Reduced VO_2_ on CPET
Hypothyroidism	Fatigue Myalgias	Abnormal TSH/FT4 Systemic metabolic/dermatologic signs
Iron-deficiency anemia	Fatigue, reduced stamina Headache	Low hemoglobin/ferritin Microcytosis
Inflammatory arthritis (RA, SLE)	Widespread pain Morning stiffness	Synovitis on exam Autoantibodies, inflammatory imaging
Small-fiber neuropathy	Burning pain Allodynia	Reduced IENFD (skin biopsy) Corneal confocal microscopy
Obstructive sleep apnea (OSA)	Non-restorative sleep Daytime fatigue	Polysomnography with apneas/hypopneas Snoring, obesity, hypertension
POTS Autonomic disorders	Orthostatic fatigue Cognitive impairment	HR increase >30 bpm on tilt Reduced HRV
IBS Functional GI disorders	Abdominal pain Altered bowel habits, fatigue	GI-predominant symptoms Rome IV criteria; exclusion of IBD
Major depression Anxiety	Fatigue Poor sleep, concentration difficulties	Core affective syndrome Psychiatric evaluation; treatment response

Abbreviations: CPET = Cardiopulmonary exercise test; FM = Fibromyalgia; FT4 = Free thyroxine; GI = Gastrointestinal; HR = Heart rate; HRV = Heart rate variability; IBD = Inflammatory bowel disease; IBS = Irritable bowel syndrome; IENFD = Intraepidermal nerve-fiber density; ME/CFS = Myalgic encephalomyelitis/chronic fatigue syndrome; OSA = Obstructive sleep apnea; POTS = Postural orthostatic tachycardia syndrome; RA = Rheumatoid arthritis; SLE = Systemic lupus erythematosus; TSH = Thyroid-stimulating hormone; VO_2_ = Oxygen consumption.

**Table 2 ijms-26-09762-t002:** Core clinical features, pathophysiological axes, diagnostic frameworks, therapeutic approaches, and future directions in fibromyalgia.

Aspect	Description and Challenges
Clinical features	Widespread pain, fatigue, and non-restorative sleep Cognitive symptoms (*fibro fog*), reduced quality of life Phenotypic heterogeneity with variable severity and course
Pathophysiology Complex multi-axis interplay with limited large-scale validation	Sensitization with altered pain network connectivity Peripheral involvement, including small-fiber pathology and mitochondrial impairment Low-grade immune activation, dysautonomia, and sleep disruption
Diagnosis	1990 tender-point criteria → 2010/2011 revisions → 2016 update Widespread Pain Index (WPI) and Symptom Severity (SS) scale integrated with clinical judgment No objective biomarkers and risk of misclassification
Treatment Partial efficacy, adherence challenges, and side effects	Pharmacological: duloxetine, pregabalin, milnacipran Non-pharmacological: cognitive behavioral therapy, graded exercise, mind–body therapies Emerging/repurposed: low-dose naltrexone, sublingual cyclobenzaprine
Future directions	Biomarker discovery via multi-omics integration Transcriptomic signature matching and network pharmacology Repurposing pipelines anchored in a mechanistic rationale Need for larger cohorts, standardization, and validation

**Table 3 ijms-26-09762-t003:** Major pathogenetic axes in fibromyalgia with representative findings and associated clinical features.

Axis	Representative Findings	Clinical Correlates
Muscle and mitochondria	Ultrastructural alterations Oxidative/nitrosative stress Impaired phosphate recovery; altered metabolites	Exercise intolerance, exertional pain Fatigue, localized tenderness Post-exertional symptom exacerbation
DRG and small fibers	Reduced intraepidermal nerve-fiber density Inflammatory sensitization of primary afferents	Burning pain, allodynia Sensory gain, widespread tenderness
Sensitization	Hyper-responsivity in insula, ACC, PFC Altered connectivity; reduced descending inhibition	Amplified pain Hyperalgesia, temporal summation
Immune and neuroinflammation	Altered cytokine/chemokine profiles Glial activation on PET	Fatigue, flu-like malaise Symptom variability, pain flares
Autonomic and sleep axes	Sympathetic predominance; reduced HRV Reduced slow-wave sleep; alpha intrusion	Morning fatigue, brain fog Lowered pain thresholds

Abbreviations: ACC = anterior cingulate cortex; DRG = dorsal root ganglia; HRV = heart rate variability; PET = positron emission tomography; PFC = prefrontal cortex.

**Table 4 ijms-26-09762-t004:** Genetic, epigenetic, and retroviral alterations implicated in fibromyalgia: pathways, evidence, and clinical implications.

Molecule/Element	Pathway/Evidence in FM	Clinical/Pathophysiological Implications
*COMT* polymorphisms	Catecholamine metabolism, altered enzymatic activity	Pain sensitivity Stress response, autonomic imbalance
*SLC6A4* (5-HTT) variants	Serotonin reuptake, association with serotonergic tone	Pain perception Mood regulation, sleep architecture
*SLC6A4*, *GRIA4*	Glutamatergic neurotransmission, linked to excitatory signaling	Central sensitization Hyperalgesia
DNA methylation (immune/oxidative loci)	Epigenetic regulation of immune and stress genes	Immune tone Redox imbalance, persistent fatigue
*GRM2* methylation	Hypermethylation reported in sibling-controlled cohorts	Potential protective effect Links glutamatergic regulation to FM risk
HERV expression profiles	Retroviral element activation; PBMC expression discriminates FM, ME/CFS, controls	Biomarker potential Epigenetic derepression may contribute to immune dysregulation
let-7 family (let-7a/d)	Nociceptor excitability, immune regulation	Small-fiber pathology Nociceptive sensitization
miR-21-5p	Inflammation, TGF-β remodeling	Tissue remodeling Chronic inflammation
miR-146a-5p, miR-155-5p	NF-κB, JAK–STAT feedback	Innate immunity dysregulation Inflammatory tone
miR-27a/b, miR-223-3p	Lipid metabolism, myeloid activation	Metabolic stress Immune activation
miR-103/107, miR-320a	Insulin signaling, mitochondrial coupling	Energetic imbalance Metabolic fatigue
miR-132/212, miR-124-3p, miR-9-5p	Synaptic plasticity, neurotransmission	Central hyperexcitability Sleep and cognitive symptoms
miR-34a-5p	Mitochondrial stress, apoptosis	Neurodegeneration Muscle fatigue

**Table 5 ijms-26-09762-t005:** Therapeutic strategies in fibromyalgia and future directions: approved, adjunctive, and emerging treatments.

Category	Description and Examples
Pharmacological (approved)	SNRIs (duloxetine, milnacipran): enhance descending inhibition and improve mood/sleep Gabapentinoids (pregabalin): reduce presynaptic glutamate release via $\alpha_{2}\delta$ subunit Low-dose tricyclics/cyclobenzaprine: mixed monoaminergic and sedative effects; may improve sleep continuity Chronic opioids: discouraged due to limited efficacy and hyperalgesia risk
Non-pharmacological (core care)	Graded aerobic and resistance exercise, sleep hygiene, CBT and acceptance-based therapies Mind–body interventions (tai chi, yoga, mindfulness): supportive for fatigue and sleep
Emerging/repurposed	Low-dose naltrexone: attenuates microglial activation via TLR4 antagonism; preliminary trials suggest benefit Ketamine: NMDA receptor antagonism; targets glutamatergic hyperexcitability, but effects are often short-lived Cannabinoid-based agents (e.g., nabilone): modest effects on sleep and pain; tolerability considerations Adjunctive strategies: vagal tone enhancement via aerobic training, sleep consolidation, pharmacological adjuncts
Future directions	Biomarker-guided stratification using multi-analyte panels (cytokines, metabolites, imaging markers) Endotype-oriented adaptive trials with early pharmacodynamic readouts (insular glutamate, cytokine shifts) Integration of gene–environment models, omics-guided repurposing, and personalized therapeutic algorithms

## Data Availability

No new data were created or analyzed in this study. Data sharing is not applicable to this article.

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
