# Peer review of "In Search of Molecular Correlates of Fibromyalgia: The Quest for Objective Diagnosis and Effective Treatments"

_ijms, 2025, doi:10.3390/ijms26199762_

Round 1
Reviewer 1 Report
Comments and Suggestions for Authors
This review addresses important and timely topics about objective diagnosis and effective treatment of Fibromyalgia (FM). These authors emphasize the importance of etiological understanding of FM, i.e. multiple omics approaches. They insist that we should move from symptom analogies to mechanism-based stratification; from “one-size-fit-all” approach to biomarker-guided precision medicine etc.
These claims may provide important implications for all researchers and healthcare professionals involved in the research and treatment of FM and will be of great help in how to proceed with future research and the development of treatments.
Minor edits and formatting:
1. There is a lot of repetition and duplication in the text, especially in sections 3, 4and 5, so it is desirable to write a little more concisely. It would be easier for readers to understand if there was a table or diagram that clearly shows the contents of Section 3 at a glance. Figure 3 seems to be too abstract to help the readers.
2. There are some citations that do not fit the content of the description. It is necessary to double-check the content of all the citations. For example, in lines 87-88 (citations for therapy of FM), paper 73 (Uceyler et al.) does not deal with therapy but small fiber pathology in FM patients, and in lines 502-504 (citations for non-pharmacological therapy), paper 73 (Uceyler et al.) does not describe therapy, but it describes pathological findings, and paper 86 (Lee et al.) mainly discusses pharmacological therapy.
3. In the section 4.1 and 4.2, Unnecessary hyphens are recognized, such as “neu- ropeptide”, ”Ke- tamine”, “moti- vates”, “sig- nature” etc.
Author Response
Comment 1. “There is repetition in Sections 3, 4, and 5; Section 3 would benefit from a summary table/diagram; Figure 3 is too abstract.”
Response. We carefully streamlined the text in Sections 3–5, removing redundancies and condensing enumerations into summary elements. Specifically:
- We introduced a new schematic overview (now Figure 2) that integrates clinical domains with molecular correlates and pathogenic axes (neuro-immune, glutamatergic, monoaminergic, energy–redox, sleep–autonomic). This replaces the previous abstract figure and allows readers to grasp Section 3 at a glance.
- We added summary tables to replace textual duplication: Table 3 (“Pathogenic axes and modes of assessment”) and Table 4 (“Genetic, epigenetic and retroviral alterations”), both of which condense Section 3 content.
- In the therapeutic section, we introduced Table 5, which clearly separates approved pharmacological, core non-pharmacological, emerging/repurposed, and future directions, thereby avoiding overlap between Sections 4 and 5.
- We restyled the Figure 1 in order to make it clearer and more informative.
Comment 2. “Some citations do not match the described content (e.g., therapy citations in lines 87–88; non-pharmacological therapies in lines 502–504; Uçeyler reference is pathology, not therapy; Lee et al. is pharmacological).”
Response. We re-aligned the references throughout:
- In the original version, non-pharmacological therapies were supported by an overly broad set of references, some of which were not pertinent (e.g., Uçeyler).
- In the revised version, non-pharmacological therapies now cite only meta-analyses and systematic reviews directly addressing exercise, CBT, and multicomponent programs (Häuser 2010; Busch 2007; Bernardy 2010).
- The Uçeyler paper was removed from the therapeutic context and is now cited appropriately in the section on small fiber pathology.
- Lee & Song is cited only in the pharmacological section (network meta-analysis), as originally intended.
This correction ensures that all references accurately match the claims made.
Comment 3. “Unnecessary hyphens in Sections 4.1 and 4.2 (‘neu- ropeptide’, ‘Ke- tamine’, ‘moti- vates’, ‘sig- nature’, etc.).”
Response. These were artifacts of line-breaking in the draft submission. We performed a thorough global revision of hyphenation and micro-editing. All terms are now correctly formatted (“neuropeptide”, “ketamine”, “motivates”, “signature”, etc.).
Reviewer 2 Report
Comments and Suggestions for Authors
The manuscript is well-structured, with an introduction that clearly outlines the relevant issues and provides all necessary background information. The review is engaging and pertinent, covering fibromyalgia in a comprehensive manner. The organization into topics and subtopics enhances clarity and understanding. However, it is worth noting that there have been recent reviews on the same subject.
Since this review aims to provide a broad overview of available information on fibromyalgia, including aspects of personalized medicine, there are several points that warrant further examination.
- The authors note that fibromyalgia (FM) prevalence is primarily seen in females, yet they do not address this topic in their discussion. While there are existing articles that explore this issue, it would be beneficial for the authors to include information regarding the differences in prevalence between males and females.
2. FM is known to overlap with several other conditions. It would be beneficial to create a table or similar format to compare the signs and symptoms of these various conditions. This approach could help identify the specific characteristics that define fibromyalgia.
3. The article emphasizes omics approaches in fibromyalgia, which I believe is one of its strengths. However, it lacks depth in discussing genetic and epigenetic aspects. Several other published articles offer more comprehensive information on these topics. To better integrate personalized medicine, it's important for this information to be aligned. Creating tables or figures could help clarify the complex genetic network involved in fibromyalgia. I recommend reviewing these topics, and there are some articles published in 2025 that may be particularly helpful.
4. Even though the references are relevant to the subject addressed in the manuscript and are mostly from 2010 to 2024, there are some references from 2024 and 2025 that could be consulted, such as those described below.
Ablin JN. Fibromyalgia: are you a genetic/environmental disease? Pain Rep. 2025 Apr 18;10(3):e1256. doi: 10.1097/PR9.0000000000001256.
Al Sharie S, Varga SJ, Al-Husinat L, Sarzi-Puttini P, Araydah M, Bal'awi BR, Varrassi G. Unraveling the Complex Web of Fibromyalgia: A Narrative Review. Medicina (Kaunas). 2024 Feb 4;60(2):272. doi: 10.3390/medicina60020272.
García-Domínguez M. Fibromyalgia and Inflammation: Unrevealing the Connection. Cells. 2025 Feb 13;14(4):271. doi: 10.3390/cells14040271.
Iannuccelli C, Favretti M, Dolcini G, Di Carlo M, Pellegrino G, Bazzichi L, Atzeni F, Lucini D, Varassi G, Leoni MLG, Fornasari DMM, Conti F, Salaffi F, Sarzi-Puttini P, Di Franco M. Fibromyalgia: one year in review 2025. Clin Exp Rheumatol. 2025 Jun;43(6):957-969. doi: 10.55563/clinexprheumatol/buhd2z.
Filipovic T, Filipović A, Nikolic D, Gimigliano F, Stevanov J, Hrkovic M, Bosanac I. Fibromyalgia: Understanding, Diagnosis and Modern Approaches to Treatment. J Clin Med. 2025 Feb 2;14(3):955. doi: 10.3390/jcm14030955.
Sedda S, Cadoni MPL, Medici S, Aiello E, Erre GL, Nivoli AM, Carru C, Coradduzza D. Fibromyalgia, Depression, and Autoimmune Disorders: An Interconnected Web of Inflammation. Biomedicines. 2025 Feb 18;13(2):503. doi: 10.3390/biomedicines13020503.
Author Response
Comment 1. “The authors note that fibromyalgia (FM) prevalence is primarily seen in females, yet they do not address this topic in their discussion. While there are existing articles that explore this issue, it would be beneficial for the authors to include information regarding the differences in prevalence between males and females.”
Response. We added a dedicated paragraph in the Discussion addressing sex differences. We highlight that the female-to-male ratio, historically ~9:1, is closer to 3:1 when applying modern diagnostic criteria. We also discuss potential biological, psychosocial, and sociocultural explanations for this discrepancy.
Comment 2. “FM is known to overlap with several other conditions. It would be beneficial to create a table or similar format to compare the signs and symptoms of these various conditions. This approach could help identify the specific characteristics that define fibromyalgia.”
Response. We incorporated a new Table 1 on differential diagnosis, which compares FM with overlapping or mimicking conditions (e.g., myofascial pain syndrome, osteoarthritis, hypothyroidism, neuropathies, ME/CFS, long-COVID, IBS, sleep disorders, anxiety/depressive disorders). The table highlights “red flags,” available tools, and when to order second-level tests, thus helping to delineate FM-specific features.
Comment 3. “The article emphasizes omics approaches in fibromyalgia, which I believe is one of its strengths. However, it lacks depth in discussing genetic and epigenetic aspects. Several other published articles offer more comprehensive information on these topics. To better integrate personalized medicine, it's important for this information to be aligned. Creating tables or figures could help clarify the complex genetic network involved in fibromyalgia. I recommend reviewing these topics, and there are some articles published in 2025 that may be particularly helpful.”
Response. We expanded Section 3 with a dedicated subsection on genetics, epigenetics, and endogenous retroviruses (HERVs). A new Table 4 summarizes loci and candidate genes (SLC6A4, SLC6A2, COMT, MAOA/B, HTR2A, DRD4, BDNF, etc.), methylation patterns (especially immuno-redox), and HERV-related findings (including the 2025 study distinguishing ME/CFS from FM). This is integrated into a broader network perspective with GSEA-based modular coherence.
Additionally, the new Figure 2 provides a visual synthesis of molecular correlates and pathogenic axes downstream of omics data.
Comment 4. “References should be updated with 2024–2025 literature (Ablin 2025; Al Sharie 2024; García-Domínguez 2025; Iannuccelli 2025; Filipović 2025; Sedda 2025).”
Response. We integrated several 2024–2025 references directly relevant to the scope of the review:
- Ablin 2025 (Pain Rep): gene–environment interactions in FM.
- Iannuccelli et al. 2025 (Clin Exp Rheumatol): state-of-the-art review.
- García-Domínguez 2025 (Cells): inflammation-FM connection.
- Filipović et al. 2025 (J Clin Med): updated diagnostic and therapeutic approaches.
- Sedda et al. 2025 (Biomedicines): FM–depression–autoimmunity network.
- We also cited the 2025 HERV paper (Biomedicines) for its relevance to molecular stratification.
We considered Al Sharie 2024 (Medicina) but noted that its narrative content overlaps substantially with other 2024 narrative reviews already included. To maintain conciseness and avoid redundancy, we did not add it, but we remain open to including it upon the Editor’s request.